# Zinc transporter SLC39A7 relieves zinc deficiency to suppress alternative macrophage activation and impairment of phagocytosis

**Wenyan Xie**⊙, **Qinghua Xue**⊙, **Liangfei Niu, Ka-Wing Wong**📧*

Shanghai Public Health Clinical Center, Fudan University, Shanghai, China

⊙ These authors contributed equally to this work.
* kwwong@gmail.com

**Data Availability Statement:** All relevant data are within the manuscript and its Supporting Information files.

## Abstract

Macrophages are key phagocytic cells and play an important role in eliminating external microorganisms and endogenous danger signals. Dysregulation in macrophage functions have been reported in patients with asthma. Zinc homeostasis is critical in maintaining macrophage functions. The solute carrier (SLC) protein SLC39A7, a Zn2+ importer, has recently been linked to asthma. However, the roles of SLC39A7 in macrophage phagocytosis are not well understood. Here we found that phagocytosis efficiency was significantly decreased in SLC39A7-knockdown THP-1 cells, however the phagocytosis capability could be reversed with zinc supplementation. SLC39A7 deficiency skewed macrophages towards alternative activation, as indicated by increased expression of M2 activation marker *CD206* and decreased expression of M1 activation marker *NOS2*. Consistent to this result, SLC39A7-knockdown cells produced reduced amounts of proinflammatory cytokines TNF- and IL-6. Furthermore, the mRNA level of receptor Clec4e previously known to be involved in phagocytosis of BCG was significantly reduced in SLC39A7 knockdown cells. Importantly, all these defects due to SLC39A7 deficiency could be reversed by zinc supplementation. Thus, zinc transporter SLC39A7 provide support for phagocytosis and classical macrophage activation.

## Introduction

Macrophages are vital first-line cells in host defense. To accomplish this task macrophages possess three critical functions: phagocytosis, antigen presentation, and immune regulation. They are essential for maintaining a normal immune state under various pathophysiological conditions [1]. Zinc homeostasis was found to be important for the innate immune system, especially for maintaining the function of macrophages [2]. Zinc plays an important role in monocyte adhesion, migration and differentiation into tissue macrophages [3], and phagocytosis of macrophages and its cytokine production [4]. Zinc supplementation increases the phagocytosis of *E. coli* and *Staphylococcus aureus* by peritoneal macrophages in a mouse model of polymicrobial sepsis [5], and inhibits alternative macrophage polarization [6].

**Funding:** This work was supported a grant to K.-W. Wong by National Natural Science Foundation of China (Award Number: 81770010). The funder had no role in study design, data collection and analysis, decision to publish, or preparation of the manuscript.

Insufficient zinc levels in cells can lead to excessive inflammatory response, producing excess cytokines and chemokines such as TNF-α, IL-6, IL-8 and IL-10 [7].

Homeostasis of intracellular zinc mainly depends on two families of zinc transporters: SLC39A/ZIP and SLC30A/ZnT. The SLC39A family members facilitate zinc influx to cytoplasm from extracellular spaces as well as intracellular compartments such as endoplasmic reticulum (ER) [8], whereas the SLC30A family members transport zinc in the opposite direction away from cytoplasm [9]. SLC39A7 is the only known ZIP member that is localized to the ER membrane [10] and is essential for regulation of cytosolic zinc levels [11]. Deletion of the SLC39A7 gene in mesenchymal stem cells leads to the accumulation of zinc in the ER, triggering overexpression of the UPR gene and endoplasmic reticulum stress [12]. Recent data shows that SLC39A7 is implicated in glucose metabolism and glycemic control in skeletal muscle cells [13, 14]. For immune cells, SLC39A7 is essential for B cell development and BCR signaling [15]. Although SLC39A7 is shown to regulate the immune system, the effect of SLC39A7 on the phagocytosis and macrophage activation is poorly understood. A transcriptional profile study identified *SLC39A7* as gene product associated with asthma [16]. Asthma is associated with impaired macrophage phagocytosis, alternative macrophage differentiation and zinc deficiency. BCG is a vaccine to prevent tuberculosis and has also been used as a potent immunomodulator in the decades [17]. BCG has been reported that the protection against asthma [18]. According to these reports, we hypothesized SLC39A7 played a critical role in macrophage phagocytosis of BCG. We generated SLC39A7-knockdown THP-1 cell lines by using CRISPR-Cas9 gene editing system. Our results revealed phagocytosis impairment and alternative macrophage polarization in SLC39A7-knockdown THP-1 cells. And importantly, these defects could be rescued with Zn2+ supplementation.

# Results

## 1. Expression of SLC39A7 increased in BCG infected macrophages

We first determined whether expression of SLC39A7 was regulated in macrophages during infection of *BCG Pasteur* stain (BCG-p). THP-1 cells were differentiated into macrophages by PMA, and then infected with BCG at a multiple of 5 (MOI). Western blotting and quantitative PCR were performed to detect the expression of SLC39A7 after infection at 6h and 24h. Both of mRNA and protein level of SLC39A7 were significantly increased in BCG-p infected THP-1 cells compared with uninfected macrophages (Fig 1A and 1B and S1 Fig).

## 2. Knockdown of SLC39A7 reduced the proliferation of THP-1 cells

To evaluate the function of SLC39A7 during infection in macrophages, we used the CRISPR-Cas9 gene editing system to generate SLC39A7-knockdown cell lines. Western blot with anti-SLC39A7 antibody showed that expression of SLC39A7 protein was abolished in

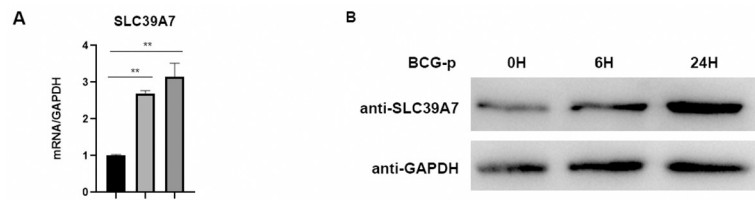

**Fig 1. SLC39A7 was up-regulated in macrophages in response to BCG-p stimulation.** The mRNA (A) and protein (B) of SLC39A7 was measured at 0 h, 6 h, 24 h after infection with BCG-p (MOI 5:1) in THP-1 cells; the experiment was performed three times. (A) was analyzed by one-way ANOVA, $^{**}$p < 0.005.

SLC39A7-knockdown cell lines (Fig 2A and S2 Fig). Cell proliferation after 4d as measured by CCK8 assay was significantly lower in SLC39A7-knockdown cells (KD-2 and KD-4) than in non-target transfected control cells (NC) (Fig 2B). Adherence rate of the SLC39A7-knock-down cells was lower than that of NC after PMA stimulation (Fig 2D). Furthermore, we determined whether supplementation of exogenous $Zn^{2+}$ could reverse the adherence defect. The result showed that the addition of $ZnCl_2$ and pyrithione, an ionophore that transports zinc through cell membrane, rescued the adhesion defect at 72 h after PMA stimulation (Fig 2D). Cell survival rates, measured at 72 h after PMA stimulation by nucleic acid stain SYTOX green that only stained DNA in dead cells, were indistinguishable between knockdown and control cells (Fig 2C). Thus, SLC39A7 knockdown reduced the rate of proliferation and adhesion by PMA-stimulated THP-1 cells.

## 3. Knockdown of SLC39A7 disrupted phagocytosis of BCG-p by THP-1 cells

Phagocytosis to pathogens by macrophages is a key innate immune effector mechanism. To evaluate the role of SLC39A7 in the host response to BCG-p infection we examined the phago-cytic capacity of SLC39A7 knockdown cells and non-targeting control (NC). Differentiated THP-1 were infected with GFP-expressing BCG-p for 4 h at a multiplicity of infection (MOI) of 5. We analysis the phagocytosis to BCG-p in THP-1 cells through fluorescence microscopy, flow cytometry, and colony-forming units (CFU) (Fig 3). Compared with NC, phagocytosis efficiency by SLC39A7-knockdown THP-1 cells was dramatically decreased (Fig 3A and 3B). CFUs in SLC39A7-knockdown cells were also lowered relative to the NC cells (Fig 3C). These results showed that the phagocytosis capacity to BCG-p was significantly reduced in SLC39A7-knockdown THP-1 cells.

Since SLC39A7 knockdown is known to reduce intracellular concentration of $Zn^{2+}$ [11] we examined intracellular Zn2+ concentration using Fluozin-3™ AM and found that, as shown

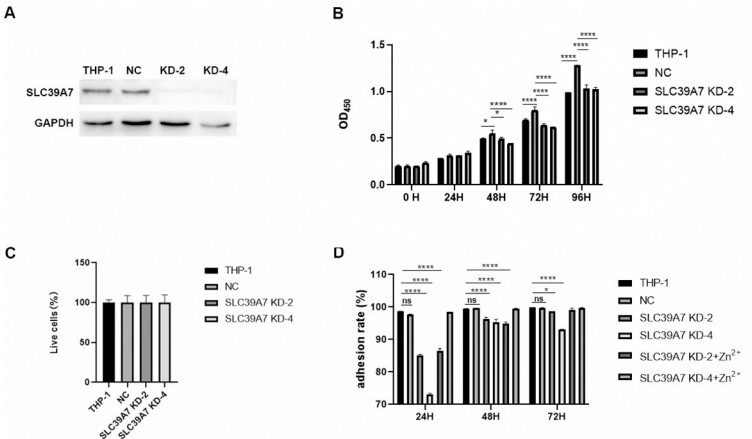

**Fig 2. Knockdown of SLC39A7 reduced the proliferation of THP-1 cells.** (A) Western blot of SLC39A7 knockdown cell lines. Equal amount of THP-1 control and SLC39A7 knockdown cell lysates were probed with anti-SLC39A7 antibody (Proteintech). (B) Proliferation of two SLC39A7 knockdown cell lines was quantitated by CCK8 assay. The result was analyzed by two way ANOVA, $^{***}p < 0.001$ compared to control (NC). (C) The percentage of living cells was determined from SYTOX Green staining under fluorescence microscopy. The cells was stained by 10 nM SYTOX Green at 72 h after PMA stimulation for 15min, then fixed by 4% paraformaldehyde, the nucleus were stained by DAPI before observing using fluorescence microscopy. (D) The cell adherence rates of SLC39A7 knockdown cells and control (NC) following stimulation with 100nM PMA for 24, 48 and 72h. SLC39A7-knockdown macrophages were treated with 5 μM ZnCl2 and 0.5 μM pyrithione. Experiments were repeated three times with similar observations, and representative data was shown. The data was analyzed by two-way ANOVA, $^{*}P < 0.05$, $^{**}p < 0.005$, $^{***}p < 0.001$, $^{****}P < 0.0005$, ns: not significant.

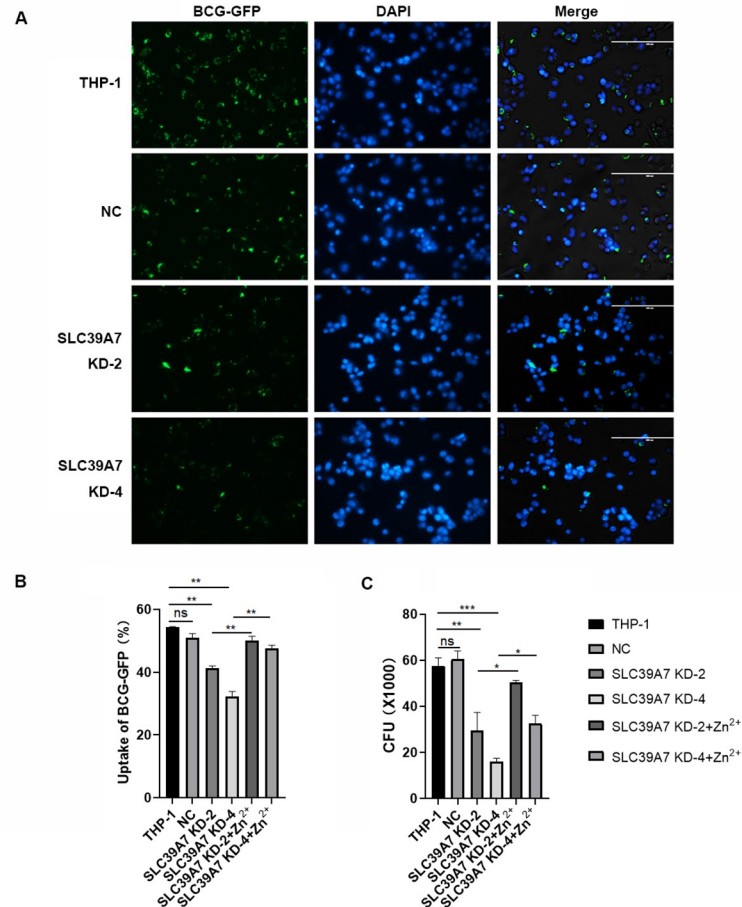

**Fig 3. The efficiency of phagocytosis to BCG-GFP in SLC39A7 KD THP-1 cells.** (A) Images from GFP-expressing BCG-p (green) infected THP-1 macrophages, the nuclei (blue) were stained using DAPI. (B) Phagocytosis of GFP-expressing BCG-p in SLC39A7 knockdown cells or control (NC). GFP-expressing BCG infected SLC39A7 KDs or NC were measured by flow cytometry. (C) Colony-forming units (CFU) of BCG-p at 3 h. (B) and (C) were analyzed by one-way ANOVA, **$P < 0.005$, ***$p < 0.001$, ****$P < 0.0005$.

in previous report, the intracellular $Zn^{2+}$ level in SLC39A7 knockdown cells was lower in SLC39A7-knockdown cells than in control cells (S3 Fig). To determine whether Zn2+ supplement could rescue the phagocytosis defect of BCG-p by the SLC39A7-knockdown THP-1 cells we found that supplementation of Zn2+ ionophore pyrithione (0.5 μM) and $ZnCl_2$ (5 μM) significantly improved the phagocytic capacity of SLC39A7-knockdown cells (Fig 3B). This result was confirmed by measuring the level of intracellular viable BCG-p (Fig 3C). Our results therefore indicated that SLC39A7 could play a role in the macrophages phagocytosis.

## 4. The mRNA expression of markers and cytokines associated with M1/M2 macrophage activation in *SLC39A7*-knockdown THP-1 cells

To study the role of SLC39A7 in macrophage phagocytosis we analyzed the mRNA expression of macrophage genes associated with classical M1 activation (*NOS2* and *CD11c*) and alternative M2 activation (*Arg-1*, *Ym-1*, and *CD206*). mRNA expression of M2 macrophage genes *Arg-1* and *Ym-1* and M1 genes *CD11c* was either unchanged in knockdown cells or unresponsive to zinc supplementation (Fig 4A and 4B). However, expression of M2 macrophage gene marker *CD206* was up-regulated in one of the SLC39A7-knockdown cells (KD4) and this up-

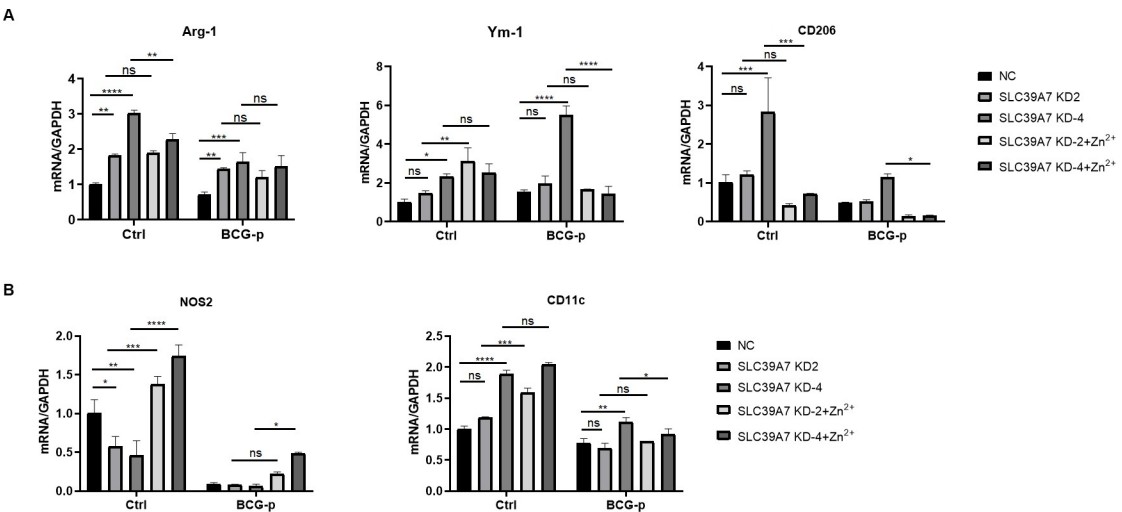

**Fig 4. The mRNA expression of M1/M2 markers in SLC39A7 KD THP-1 cells.** The mRNA Level of Arg-1, Ym-1, CD206 (A), NOS2, CD11C (B) in knockdown cells or control response to BCG-p infection at 6 h (MOI = 10). Data was representative of three independent experiments and expressed as mean ± SD for three biological replicates. All data was analyzed by two-way ANOVA, *P < 0.05, **P < 0.005, ***p < 0.001, ****P < 0.0005, ns: not significant.

regulation could be reversed by zinc supplementation (Fig 4A). Expression of M1 marker *NOS2* was down-regulated in SLC39A7-knockdown cells and this down-regulation was reversible by zinc supplementation (Fig 4B).

Classically activated M1 macrophage has a proinflammatory profile dominated by cytokines TNF-α and IL-6, whereas alternatively activated M2 macrophage has an anti-inflammatory profile dominated by IL-10. Although no evidences supporting a role for SLC39A7 in regulating expressions of *TNF-α*, *IL-6*, and *IL-10* in a fashion reversible by zinc supplementation were found (Fig 5A), secretion of TNF-α and IL-6 proteins at 48h after BCG-p infection

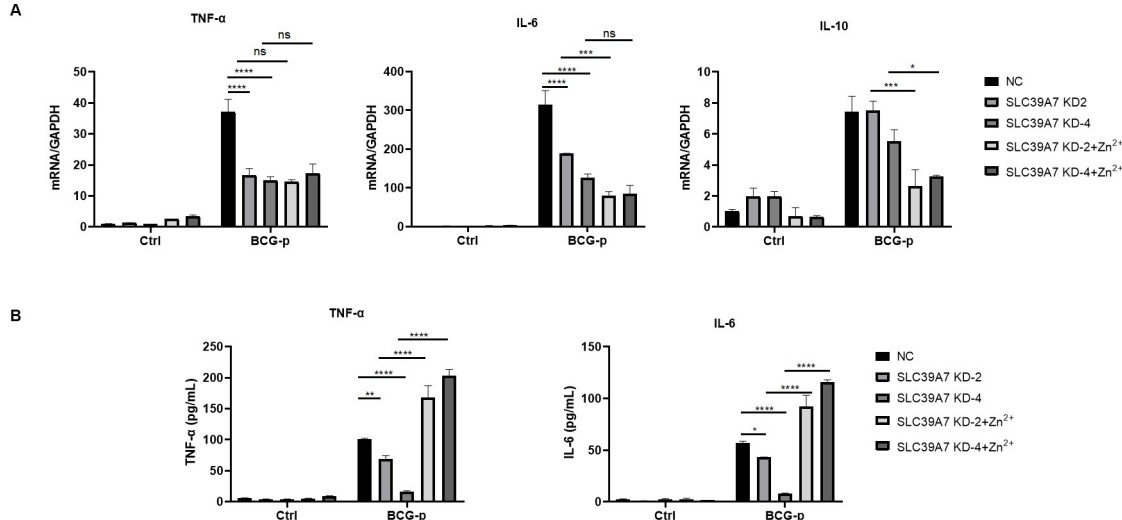

**Fig 5. The cytokines expression in SLC39A7 KD THP-1 cells.** the mRNA levels of TNF-α, IL-6 and IL-10 were tested in knockdown and control cells at 6 h after BCG-p infection (A), and the concentrations of TNF-α and IL-6 in the supernatants were examined at 48h after BCG-p infection by ELISA kits according the manufacturer's protocol (NeoBioscience) (B). The data was analyzed by two-way ANOVA, *P < 0.05, **P < 0.005, ***p < 0.001, ****P < 0.0005, NS: not significant.

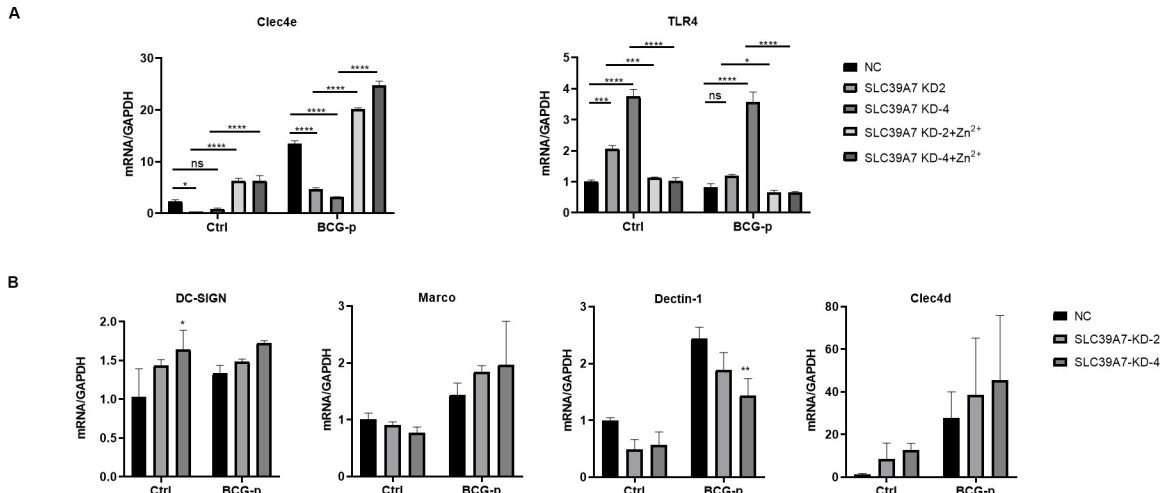

**Fig 6. The mRNA expression of cell surface receptors in SLC39A7 KD THP-1 cells.** The mRNA of Clec4e, TLR4, DC-SIGN, MARCO, Dectin-1, Clec4d in knockdown cell or control response to BCG-p infection at 6 h (MOI = 10). All data was analyzed by two-way ANOVA, $^*P < 0.05$, $^{**}P < 0.005$, $^{***}p < 0.001$, $^{****}P < 0.0005$, NS: not significant.

were reduced in SLC39A7-knockdown cells than in control cell and this reduction in secretion of TNF-α and IL-6 was reversible by zinc supplementation (Fig 5B). IL-10 protein was undetectable at 48h after BCG-p infection. Taken together, our results indicate that SLC39A7 likely play a role in classical M1 activation of macrophages.

## 5. The expression of *Clec4e* was reduced in SLC39A7-knockdown cells

Bacteria use their cell surface associated macromolecular structures to interact with specific receptor molecules on host cells. We therefore examined the mRNA expression of cell surface receptors previously known to be involved in phagocytosis. The results showed that the mRNA level of receptors *Clec4e* (also known as Mincle) was significantly decreased and *TLR4* mRNA level was increased in SLC39A7 knockdown cells (Fig 6). However, *Clec4e* expression increased significantly and *TLR4* expression decreased when exogenous $Zn^{2+}$ was added in the knockdown cells. In contrast, the mRNA levels of other surface receptors DC-SIGN, MARCO, Dectin-1 and Clec4d were comparable with that in control. The deficiency of SLC39A7 could reduce the transcription level of Clec4e.

## Discussion

Macrophages are critical first responders. They are specialized to recognize and eliminate pathogens [19]. The level of intracellular zinc is known to influence the phagocytosis capacity of macrophages [20]. SLC39A7, the only known SLC39A family member localized on the ER membrane [10], is essential for regulation of cytosolic zinc level [11]. Previous studies have examined the role of SLC39A7 on Notch [21] and insulin signaling [22–24]. However, the role of SLC39A7 in phagocytosis has not been previously examined. In this report, we generated SLC39A7 knockdown THP-1 cell lines by using CRISPR-Cas9 gene editing system. Our results showed that phagocytosis was severely inhibited in the SLC39A7 knockdown THP-1 cells. Importantly, this defect in phagocytosis could be restored by supplementation of exogenous $Zn^{2+}$. Thus, SLC39A7 plays a role in the phagocytosis of BCG by THP-1 cells.

Macrophages possess a broad array of cell surface receptors, intracellular mediators and essential secretory molecules for recognition, engulfment and destruction of invading

pathogens and also for regulation of other kinds of immune cells [25]. The C-type lectin receptor Clec4e plays a role in eliciting inflammatory response against mycobacterium and mediates neutrophil phagocytosis and extracellular trap formation [26]. We found that the expression of *Clec4e* was dramatically decreased in SLC39A7 knockdown cells. This result could explain our observation that SLC39A7 knockdown negatively affected phagocytosis.

Toll-like receptors (TLRs) play a crucial function in macrophage maturation and activation. TLR4 expressed by macrophages are among the first line of defense against pathogens [27]. The expression of TLR4 was slightly increased in SLC39A7 knockdown cells. This effect could be due to increased cell death in SLC39A7 knockdown cells. The deficiency of SLC39A7 is known to cause cell death and release of the extracellular inflammatory factor HMGB1, which could activate TLR4 expression [28]. Moreover, TLR4 activation impaired phagocytic capacity of microglia [29]. It could further explain the defective phagocytosis in SLC39A7 knockdown cells.

Macrophages are highly plastic and can be differentiated into two major phenotypic subsets in response to different ambient stimulus: pro-inflammatory M1 phenotype and anti-inflammatory M2 phenotype [30]. Cellular zinc homeostasis could modulate differentiation of macrophages [31]. It is known that high Zn2+ promotes M1 differentiation while inhibits M2 differentiation, while Zn2+ deficiency inhibits the differentiation of M1. Given that SLC39A7 deficiency lowered the cytosolic Zn2+ levels, we investigated the role of SLC39A7 in the differentiation of macrophages and measured gene expression of M1/M2-associated markers and cytokines. In accordance with previous studies [32–34], we selected CD11c and NOS2 as markers of M1 macrophage activation, and Arg-1, Ym-1 and CD206 as indicators of M2 macrophage responses. Compared with wild type cells, the mRNA of the M2 marker *CD206* increased significantly in KD4 cells, and the expression of M1 marker *NOS2* was decreased in both KD2 and KD4 cells. Moreover the pro-inflammatory cytokines TNF-α, which induce macrophage activation and cytotoxic activity [35], was decreased in SLC39A7 knockdown cells. IL-10 is a well-known anti-inflammatory expression increased and IL-10 plays a negative role in BCG-induced macrophage cytotoxicity [36]. One of the KD cells KD-2 did not exhibit defects in several cytokine expressions (Fig 4A and 4B), and was unresponsive zinc supplementation. We presumed that the different adhesion of the two knockdown cells caused the discrepancy. KD-4 has a bigger defect in adhesion and a more responsiveness in cytokine productions. It is interesting to note that monocytes from systemic lupus erythematosus have defects in adhesion and produce elevated levels of IL-6, TNF-a, and IL-10 than healthy monocytes [37]. Thus, the discrepancy between KD-2 and KD-4 could reflect a differential compensatory response by the KD cells. In summary, our results indicated that SLC39A7 could regulate macrophage polarization and balance of pro-inflammatory and anti-inflammatory responses.

Our results might have significances towards the pathophysiology of asthma in humans. Alternative M2 macrophage activation and impaired macrophage phagocytosis were two characteristics in patients with asthma [38, 39]. Interestingly, patients with asthma are known to have significantly lower plasma levels of $Zn^{2+}$ [40]. It is possible that this $Zn^{2+}$ deficiency is responsible for the M2 macrophage activation, and impaired macrophage phagocytosis. To compensate for the $Zn^{2+}$ deficiency, *SLC39A7* is upregulated in patients with asthma than healthy controls as previously reported [16]. Recent data shows that SLC39A7 is implicated in glucose metabolism and glycemic control in skeletal muscle cells [13, 14]. SLC39A7 has essential role in B cell development, SLC39A7 was required for BCR signaling [15]. All these observations suggested that SLC39A7 played an essential role in the host immune metabolism. The function of SLC39A7 should be further investigated.

## Materials and methods

### Cell culture

The human myelogenous leukemia cell line THP-1 was purchased from ATCC (Manassas, VA). The cells were maintained in RPMI 1640 with 10% fetal bovine serum at 37˚C in a 5% $CO_2$ humidified incubator.

### Generation of Knockdown cell line with CRISPR/Cas9

Guide RNA sequences for CRISPR/Cas9 were designed at CRISPR design web site (http://crispr.mit.edu/), provided by the Feng Zhang Lab. The SLC39A7 gRNA sequences were shown as follows: gRNA2:5′ –CACCG**CTCTCCCTCACCAGGCACTG**–3′ and gRNA4: 5′ –CA CCG**AGCTGCTGAGATCAGCACTG**–3′ . The complementary oligonucleotides for gRNAs were annealed, and cloned into LentiCRISPRv2 (Addgene, Cambridge, MA). THP-1 cells were transfected with LentiCRISPRv2/gRNA using Lipofectamine 2000 (ThermoFisher, Waltham, MA, USA) following the manufacturer's recommended protocol. Two days after transfection, cells were treated with 1 μg/ml of puromycin (Sigma-Aldrich, Merck KGaA, Darmstadt, Germany) for three days. After two weeks, the single cell was isolated with Flow cytometry into 96 well plates. The knockdown efficiency was evaluated with the protein expression of SLC39A7 by western blot.

### Cell Counting Kit-8 (CCK8) assay

The cell proliferation potential of SLC39A7 knockdown was measured with CCK-8 assay (Dojindo) according to the manufacturer's instruction. In brief, the indicated cells (at a cell density of $1.5 \times 10^4$ cells /well) were seeded into a 96-well plate and cultured in normal growth medium. 10 μl CCK-8 working solution was added to each well and incubated at 37˚C for 4 h. The absorption at 450 nm was measured at indicated time points (24, 48, 72, and 96 h, respectively) using an ELISA plate reader. Experiments were performed in triplicate.

### Cell adherence assay

THP-1 cells ($3 \times 10^6$ cells/well) were seeded in 24-well plates with 100nM phorbol myristate acetate (PMA) (Sigma-Aldrich) for 0, 1, 2 and 3 days, respectively. Unattached cells in the supernatant were counted. The cell adherence rate was calculated according to the cell number of THP-1 cells without stimulation.

### Bacterial strains and cultures

The *Mtb* strain *BCG-Pasteur* (BCG-p) or BCG-p harboring the plasmid pMF42A encoding green fluorescent protein (GFP) were grown on Middlebrook 7H9 medium (Difco, Becton Dickinson, Franklin Lakes, NJ, USA) with complete supplements [10% OADC, 0.5% glycerol and 0.05% Tween-80]. BCG-p was grown to logarithmic phase or an OD600 of between 0.45 and 0.85. Before infection, bacteria were pelleted in a 2ml tube with attached loop cap, suspended in 1 ml of PBS in a 15 ml conical tube, and were sonicated three time for 15 s (80 output and 100% duty cycle). The OD600 of each strain was then measured. 1 OD600 is equal to about $3 \times 10^8$ CFU ml-1. For infection of THP-1, 10% normal human serum (Gemini, CA, USA) was used to facilitate *BCG-p* binding to host cells. After 3 h incubation at 37˚C, infected cells were washed three times gently with room temperature PBS and then maintained in RPMI 1640 containing 10% fetal bovine serum (FBS).

## Reverse transcription and quantitative real-time PCR

Total RNA was extracted using Trizol reagent (Takara), according to the manufacturer's protocol. The fist-stand cDNA was synthesized using ImProm-II Reverse Transcription System (Promega) according to the manufacturer's instructions. Quantitative PCR was performed on the Applied Biosystems 7500 Real-Time PCR System (ABI, USA) with SYBR Premix Ex Taq (Takara). Expression of each gene was normalized to the expression of the housekeeping gene, glyceraldehyde-3-phosphate dehydrogenase (GAPDH). Data was analyzed using the 2-ΔΔCt method. PCR for each sample was performed in triplicates. The sequences of primers are provided in the S1 Table.

## ELISA

The supernatants were collected at 48h after BCG-p infection. The level of TNF-α and IL-6 was determined by ELISA using ELISA kits according the manufacturer's protocol (NeoBioscience).

## Phagocytosis assay

Differentiated macrophages were infected with GFP-expressing BCG-p for 4 h at a multiplicity of infection (MOI) of 5. After the infection, cells were washed three times gently with room temperature PBS and were fixed in 4% paraformaldehyde containing culture medium for 15min at 37˚C. The nuclei were stained using DAPI. Cells were examined under fluorescence microscopy. For flow cytometry, the phagocytosis to BCG-p was performed as previously report [41]. Briefly, THP-1 cells were infected by GFP-expressing BCG-p for 4 h, then the cells were rinsed with PBS, next the cells were removed form plates and were measured with BD Accuri C6. The data was analyzed by the soft of Flowjo V10. Phagocytosis was determined by gating the cells and calculating the percent of cells that had taken up GFP-expressing BCG-p.

## Colony-forming unit assay

Macrophages were infected in vitro with BCG-p at an MOI of 1, and incubated at 37˚C in a 5% CO2 humidified incubator. After infection 3 h, the cells were washed three times with PBS to remove extracellular bacteria and continued to culture with 50 μg/mL gentamycin for 1 h. Then the cells were rinsed three times with PBS, and were lysed by PBS containing 0.5% Triton X-100. The lysates were serially diluted and spotted on agar (7H10, 10% OADC) in triplicate. Colonies were counted after 2–3 weeks.

## Statistical analysis

Experiments were repeated at least three times. All statistical analyses were performed using GraphPad Prism 8.0. The statistical comparison was evaluated with one- or two-way ANOVA. $P < 0.05$ was considered statistically significance.

## Supporting information

**S1 Fig. The raw image of protein expression of SLC39A7 in THP-1 cells.**
(TIF)

**S2 Fig. The raw image of protein expression of SLC39A7 in knockdown cells.**
(TIF)

**S3 Fig. The intracellular Zn concentration was measured using Fluozin-3$^{TM}$ AM.**
(TIF)

**S1 Table. The list of primers.**
(DOCX)

## Acknowledgments

We thank colleagues Hui Ma and Kang Wu for helpful advice.

## Author Contributions

**Conceptualization:** Wenyan Xie, Qinghua Xue, Ka-Wing Wong.

**Formal analysis:** Wenyan Xie, Qinghua Xue.

**Funding acquisition:** Ka-Wing Wong.

**Investigation:** Wenyan Xie, Qinghua Xue, Ka-Wing Wong.

**Resources:** Liangfei Niu.

**Supervision:** Ka-Wing Wong.

**Writing – original draft:** Wenyan Xie.

**Writing – review & editing:** Qinghua Xue, Ka-Wing Wong.

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
