## [Decision Letter · Decision Letter 0]

6 Mar 2020

PONE-D-20-03246

Zinc transporter SLC39A7 relieves zinc deficiency to suppress alternative macrophage activation and impairment of phagocytosis

PLOS ONE

Dear Dr. Wong,

Thank you for submitting your manuscript to PLOS ONE. After careful consideration, we feel that it has merit but does not fully meet PLOS ONE’s publication criteria as it currently stands. Therefore, we invite you to submit a revised version of the manuscript that addresses the points raised during the review process.

Both Reviewers thought that the data were interesting but picked up on many mistakes in the manuscript relating to data analysis and inconsistencies between figures and data descriptions in the main text. These all need to be addressed in the revised manuscript. 

We would appreciate receiving your revised manuscript by Apr 20 2020 11:59PM. To enhance the reproducibility of your results, we recommend that if applicable you deposit your laboratory protocols in protocols.io, where a protocol can be assigned its own identifier (DOI) such that it can be cited independently in the future. For instructions see: http://journals.plos.org/plosone/s/submission-guidelines#loc-laboratory-protocols

We look forward to receiving your revised manuscript.

Kind regards,

Rebecca A Hall

Academic Editor

PLOS ONE

Journal Requirements:

"This work was supported by grants from National Natural Science Foundation (81770010)."

"NO"

Reviewers' comments:

Reviewer's Responses to Questions

**Comments to the Author**

1. Is the manuscript technically sound, and do the data support the conclusions?

Reviewer #1: Partly

Reviewer #2: Partly

2. Has the statistical analysis been performed appropriately and rigorously? 

Reviewer #1: No

Reviewer #2: Yes

3. Have the authors made all data underlying the findings in their manuscript fully available?

Reviewer #1: Yes

Reviewer #2: Yes

4. Is the manuscript presented in an intelligible fashion and written in standard English?

Reviewer #1: No

Reviewer #2: Yes

5. Review Comments to the Author

Reviewer #1: The authors describe the link between the SLC39A7 zinc transporter knockdown and its effect on phagocytosis and M1 and M2 macrophage response. While this manuscript is covering an interesting area some of the conclusions are not in line with the results shown and the link between asthma and BCG is not discussed.

The manuscript has many English mistakes and is not fully formatted (1. Headline missing number, 5. Headline still includes ???). Some language is non-scientific (line 97 – partially rescue the situation). The statistics are only partially done and the p-values in the legend don’t match the ones show in the figures. The quality of the figures is rather poor with very low resolution which make the axis labels blurry. There are two supplementary figures which are not mentioned anywhere.

Explain why mRNA experiments were done with MOI of 10 when all other experiments were done with MOI of 5?

Figure1: No statistics on time point 24h, no indication of how many repeats were done, for p-values only ** and *** are mentioned in the legend which actually are not shown in the figure, the increase on SLC39A7 on protein level is not that obvious, a quantification of signal strength with image J would be beneficial.

Figure2: The decrease of proliferation is rather slow and there is no difference for the first 72h and only a small but significant increase after 96h. And yet adhesion studies omit this time point and only go until 72h. It is claimed that the addition of Zn and pyrithione rescues the failure in adhesion which can only be shown for 1 of the 2 SLC39A7 knockdown (KD-4) whereas KD-2 is unresponsive to the addition of zinc. There is mentioning of all THREE knockdown cell lines in figure legend 2 when only 2 were analysed. Statistic don’t match up again.

Figure3: No mentioning of sonicated vs non-sonicated anywhere in the in the text or legend (either take out the of the figures or explain in text).

Figure4: Shows the same before mentioned discrepancy between the two different knockdown cell lines. M2 marker Arg-1 was only increased in one of the two KDs. The authors picked the one that fits his theory best and did not discuss the unresponsiveness of KD-2. Same goes for IL-10 expression. Stats, p-values don’t match again. Stats, p-value

Figure5: TLR4 is already up before infection and is increased upon BCG phagocytosis. Address this phenomenon. Stats, p-value

Figure6: How many repeats?

Especially due to the different responses of the two KD cell lines this paper would benefit from measuring the actual zinc level of the knockdowns compared to NC by either ICP-MS/ICP-OES or by FluoZin (or similar) and flow cytometry. I also suggest measuring interleukin level directly by ELISA rather than extrapolate their increases by gene expression.

Reviewer #2: The authors of the article 'Zinc transporter SLC39A7 relieves zinc deficiency to suppress alternative 2 macrophage activation and impairment of phagocytosis' are providing interesting observations about the role the zinc transporter SLC39A7 on macrophage functions and polarization.

This manuscript is solely descriptive and doesn't provide insight about the molecular mechanisms under the control of SLC39A7. However, I think these observations are interesting and original. They bring a new set of evidences suggesting that the zinc has an important role in modulating the immune response and the host-pathogen interactions.

I think the manuscript could be improved by the following suggestions:

Major comments:

Line 63: SLC39A7 links with asthma. The reference is missing here. Is it https://www.jimmunol.org/content/197/2/655 ? Please provide the related reference.

Fig. 2: What is the difference between THP-1 cells and NC cells? Why are the THP-1 (which are a the original non-transfected cell line) not behaving as this non-target transfected NC cells but as the KD#2 and KD#4? This point is important as it suggests that KD#2 and KD#4 are proliferating at the same rate as conventional THP-1. However, the authors don’t stress this point and prefer to focus on the results obtained with the NC strain.

Line 89, Line 92, Fig. 2 B: considering the way this CCK8 assay is set-up, I think it is incorrect to write ‘cell viability’ here as the authors are in fact measuring the cell number (which can be affect the cell proliferation rate rather than the death rate) via the CCK8 assay.

Line 98 to 100: What is the timepoint of cell culture at which the SYTOX Green staining has been performed?

Legend Figure 2 C: What is the time point of culture at which the SYTOX Green staining has been performed?

Line 117, line 118, line 121, line 124, line 138: As the authors studied the transcript expression levels of genes associated with the pro-inflammatory (M1) and resolving (M2) macrophages, and not the actual protein expression from these genes, it would be more accurate to write that they analysed ‘the transcript expression levels of genes’ rather than ‘expression of markers’, ‘the surface protein CD11c’, ‘the expression of cytokines’, ‘expression of receptors Clec4e’.

Line 133: Once again I would advised to write ‘the transcript expression levels of Clec4e was reduced…’ as the authors are just using quantitative PCR for those experiments and not the actual protein levels.

Line 142-144, I don’t think the authors can come to the conclusion that ‘These results suggested that SLC39A7 played a role in the macrophage phagocytosis through mediating the expression of cell surface protein Clec4e in sensing microbial invasion’ as they don’t show any results related to differential protein expression of CLEC4e (Mincle) in their cell lines and didn’t try to rescue the phenotype observed by forcing CLEC4e expression in those cell lines or didn’t try to mimic the phenotype observed by blocking CLEC4e in their assay.

Results, Paragraph 5: Why are the author not describing in the results section the differences they observed for TLR4 gene expression (Figure 5). It is one of their major finding that could also explain the differences observed with the phagocytosis?

Line 179: ‘The expression of TLR4’, please add ‘gene’ as you are looking for the transcript and not the protein.

Line 196: ‘the pro-inflammatory cytokines TNF-a’, please add ‘the gene coding for ’ as you are looking for the transcript and not the protein. Same thing on line 197 for IL-10 and line 200 for IL-6.

Supplemental Figure: The authors are providing two supplemental figures, however there is no mention of these figures inside the result section of the manuscript.

Material and Methods, Phagocytosis Assay section: The authors report that they have performed some of the phagocytosis assay by flow cytometry. However, there is no obvious reference about these flow cytometry experiments in the rest of the manuscript.

Minor comments:

Introduction and Discussion: I think this manuscript would benefit from citing and discussing recent important works done on SLC39A7 in B cell development (doi: 10.1038/s41590-018-0295-8) and glycolysis (doi: 10.1371/journal.pone.0079316. eCollection 2013.) in the context of immune cells crosstalk and the importance of immuno-metabolism in macrophage functions and polarization.

6. PLOS authors have the option to publish the peer review history of their article (what does this mean?). If published, this will include your full peer review and any attached files.

Reviewer #1: Yes: Claudia Simm

Reviewer #2: Yes: Guillaume E. Desanti

---

## [Author Response · Author response to Decision Letter 0]

21 Apr 2020

Point-to-point responses

1. Response to Reviewer #1

1) Q: manuscript is covering an interesting area some of the conclusions are not in line with the results shown and the link between asthma and BCG is not discussed. The manuscript has many English mistakes and is not fully formatted. 

A: We thought carefully our results and conclusion, and agreed that our conclusion was improper, so we have corrected our conclusion according our experimental results (line 43-44 and line 150-151). We agreed that we made an inaccurate conclusion: ‘These results suggested that SLC39A7 played a role in the macrophage phagocytosis through mediating the expression of cell surface protein Clec4e in sensing microbial invasion.’ This statement is not replaced with the following statement:

Line 43-44: “Taken together, these results suggested that SLC39A7 played a role in the macrophage phagocytosis.”

In the introduction section, we also add some information about the link between asthma and BCG (line 73-75). 

Line 73-75: “BCG is a vaccine to prevent tuberculosis and has also been used as a potent immunomodulator in the decades [17]. BCG has been reported that the protection against asthma [18].”

We went over our manuscript again three times and corrected many mistakes and inappropriate words. The statistics and figures were done again, and the information of supplementary figures (line 89, line 94, line 133-136, line 154-156 and the supporting information section) was added in the manuscript. 

2) Q: Explain why mRNA experiments were done with MOI of 10 when all other experiments were done with MOI of 5?

A: We performed the mRNA experiments at 6 h after BCG-p infection; we increased the MOI to 10 for a higher host response to the BCG-p. In the other experiments, the BCG-p at MOI of 5 was used to display the difference between control cell and knockdown cell better.

3) Q: Figure1: No statistics on time point 24h, no indication of how many repeats were done.

A: We did mistakenly maked these errors in the statistics due to our negligence. We have reproduced Fig.1A and corrected the indistinctive description using Graphpad Prism 8.0 (line 420).

4) Q: Figure2: The decrease of proliferation is rather slow and there is no difference for the first 72h and only a small but significant increase after 96h. And yet adhesion studies omit this time point and only go until 72h. 

A: The data of Fig.2B was analyzed again (line 426). At the 72 h in the proliferation assay, there is difference between control and SLC39A7 knockdown cell, but it was difficult to visualize the difference, we changed the graph type of Fig.2B and showed the difference between KDs and control (NC). Moreover, because all experiments were performed at 72 h after PMA stimulation, the THP-1 cells would become macrophages that incapable of proliferation, thus the adhesion assay was done only up to 72 h. We also corrected the word of ‘three’ in the figure legend (line 425). 

5) Q: Figure3: No mentioning of sonicated vs non-sonicated anywhere in the in the text or legend (either take out the of the figures or explain in text).

A: We believed the result infection of non-sonicated BCG-p in Fig.3 was irrelevant and therefore was removed in the revised verison. 

6) Q: Figure4: The authors picked the one that fits his theory best and did not discuss the unresponsiveness of KD-2. Same goes for IL-10 expression. Stats, p-values don’t match again. Stats, p-value.

A: We repeated our experiments at least three times, and got same trend. We presumed that the different adhesion of the two KD cells caused the discrepancy (line 98-100, Fig. 2D). KD-4 has a bigger defect in adhesion and a more responsiveness in cytokine productions. It is interesting to note that monocytes from systemic lupus erythematosus have defects in adhesion and produce elevated levels of IL-6, TNF-a, and IL-10 than healthy monocytes (Yi et al., 2010, Arch Immunol Ther Exp, PMID 20676786). Thus, the discrepancy between KD-2 and KD-4 could reflect a differential compensatory response by the KD cells. We also measured the intracellular concentration of Zn2+ by Fluozin-3TM AM using flow cytometry and found they were significantly decreased in the two KDs, but there was no difference between the two KDs (Figure S4). The statistics analysis was done again, and the p-value was corrected due to our careless (line 433-434).

7) Q: Figure5: TLR4 is already up before infection and is increased upon BCG phagocytosis. Address this phenomenon. Stats, p-value

A: The TLR4 signaling could be activated by extracellular inflammatory molecule HMGB1. The deficiency of SLC39A7 could induce cell death and lead to the release of HMGB1. Thus the increase of TLR4 mRNA could be induced by HMGB1 in knockdown SLC39A7 cells (line 183-186). The statistics analysis was done again, and the p-value was corrected due to our carelessness (line 449).

8) Q: Figure6: How many repeats?

 A: All experiments were done at least three times. According to reviewer’s suggestion, we measured the intracellular zinc level by Fluozin-3 AM, and added the figure in supplemental section (Figure S4, line 154-156 ), the result showed that the intracellular Zn2+ concentration in KDs was significantly lower than that in control cells. We also performed ELISA assays for interleukin level at 48h after BCG-p infection (Figure S3, line 133-136), the results showed that proinflammatory cytokines TNF-α and IL-6 were significantly decreased in KDs at 48 hour after BCG-p infection. The mRNA of IL-6 was inconsistent with the protein, which could be caused by the improper expression of mRNA. ER is the place of mRNA translation; the knockdown of SLC39A7 could induce the UPR response, which leads the improper translation of mRNA. 

2. Response to Reviewer #2

1) Q: Line 63: SLC39A7 links with asthma. The reference is missing here. Is it https://www.jimmunol.org/content/197/2/655 ? Please provide the related reference.

 A: Thanks for your kindly reminder. We added the reference in the Introduction (line 71, ref.16).

2) Q: Fig. 2: What is the difference between THP-1 cells and NC cells? However, the authors don’t stress this point and prefer to focus on the results obtained with the NC strain.

 A: NC cells are THP-1 cells transfected with control plasmid containing no target sequence. Because the KD#2 and KD#4 cells are THP-1 cells transfected with plasmid containing target sequence, we thought that we should use NC cells as control cells. The phagocytosis efficiency of the THP-1 cells and NC cells are comparable, so in the other experiments we performed assays only in NC cells. We checked the proliferation rate of all cells three times and found that the proliferation rate of NC was faster than others, but we don’t know how it caused, it may be caused by the different cleaved position by Cas9. 

3) Q: it is incorrect to write ‘cell viability’ here as the authors are in fact measuring the cell number via the CCK8 assay.

 A: We corrected the word of “cell viability” with cell proliferation rate (line 95, line 98).

4) Q: Line 98 to 100: What is the timepoint of cell culture at which the SYTOX Green staining has been performed?

 A: We checked the cell death with SYTOX Green staining at 72 h after PMA stimulation (line 103).

5) Q: Legend Figure 2 C: What is the time point of culture at which the SYTOX Green staining has been performed?

 A: We performed the SYTOX Green staining at 72 h after PMA stimulation, and added the timepoint in the figure 2C (line 427-430).

6) Q: As the authors studied the transcript expression levels of genes associated with the pro-inflammatory (M1) and resolving (M2) macrophages, it would be more accurate to write that they analysed ‘the transcript expression levels of genes’ rather than ‘expression of markers’. 

 A: We corrected the words ‘the expression of markers’ with ‘the mRNA levels of genes in all corresponsive positions (line 123, line 125, and line 126).

7) Q: Line 133: Once again I would advised to write ‘the transcript expression levels of Clec4e was reduced…’

A: We changed the word ‘expression’ with ‘the mRNA level’ (line 146). 

8) Q: Line 142-144, I don’t think the authors can come to the conclusion that ‘These results suggested that SLC39A7 played a role in the macrophage phagocytosis through mediating the expression of cell surface protein Clec4e in sensing microbial invasion’. 

 A: We have corrected our conclusion and deleted the words ‘mediation the expression of Clec4e in sensing microbial invasion’ (line 43-44, line 150-151).

Line 43-44: “Taken together, these results suggested that SLC39A7 played a role in the macrophage phagocytosis.”

Line 150-151: “These results suggested that SLC39A7 played a role in the macrophage phagocytosis; the deficiency of SLC39A7 could reduce the transcription level of Clec4e.”

9) Q: Results, Paragraph 5: Why are the author not describing in the results section the differences they observed for TLR4 gene expression (Figure 5). It is one of their major finding that could also explain the differences observed with the phagocytosis? 

A: Thank you for your kindly suggestion, we neglected the results of TLR4 gene expression due to our focus on the Clec4e gene expression. In the new manuscript, we added the analysis of TLR4 gene expression (line 146 and line 228-231).

Line 146: “The results showed that the mRNA level of receptors Clec4e (Mincle) was significantly decreased and TLR4 mRNA was increased in SLC39A7 knockdown cells (Fig.5).”

Line 183-186: “The expression of TLR4 was slightly increased in SLC39A7 knockdown cells. The phenomenon could be caused by the cell death in SLC39A7 knockdown cells. The deficiency of SLC39A7 leads to the cell death and releases the extracellular inflammatory factor HMGB1, which could activate TLR4 expression [29].”

10) Q: Line 179: ‘The expression of TLR4’, please add ‘gene’ as you are looking for the transcript and not the protein.

 A: Thank you for your advice again, we corrected ‘the expression of TLR4’ with ‘the TLR4 mRNA level’ (line 146).

11) Q: Line 196: ‘the pro-inflammatory cytokines TNF-a’, please add ‘the gene coding for ‘. Same thing on line 197 for IL-10 and line 200 for IL-6.

 A: We corrected all ‘the expression of’ with ‘the gene expression of’ in the RT-PCR results (line 129, line 130, and line 131).

12) Q: there is no mention of supplement figures inside the result section of the manuscript.

 A: We added the supplemental figures at their corresponsive positions in the manuscript (line 89, line 94, line 136, line 156, and line 291-296). 

13) Q: there is no obvious reference about these flow cytometry experiments in the rest of the manuscript.

 A: In the Figure 3B and 6A, we performed the phagocytosis assay by flow cytometry. We also added a reference (ref.42, Kordon AO, et al., 2017, Front Microbiol. doi: 10.3389/fmicb.2017.02638.) in the method of phagocytosis assay. 

14) Q: Minor comments: I think this manuscript would benefit from citing and discussing recent important works done on SLC39A7 in B cell development (doi: 10.1038/s41590-018-0295-8) and glycolysis (doi: 10.1371/journal.pone.0079316. eCollection 2013.)

A: Thank you for the suggestion. We added discussion about the SLC39A7 had reported in B cell development and glycolysis, all these results hinted the importance of SLC39A7 in immune-metabolism (line 65-68 and line 214-218). 

Line 65-68 in Introduction: “Recent data shows that SLC39A7 is implicated in glucose metabolism and glycemic control in skeletal muscle cells [13, 14]. In the immune cells, it was reported that SLC39A7 has essential role in B cell development; the intact of SLC39A7 was required for BCR signaling [15].”

Line 214-218 in Discussion: “Recent data shows that SLC39A7 is implicated in glucose metabolism and glycemic control in skeletal muscle cells [13, 14]. SLC39A7 has essential role in B cell development, SLC39A7 was required for BCR signaling [15]. All these observations showed that SLC39A7 played an essential role in the host immune metabolism. The function of SLC39A7 should be further investigated.”

---

## [Editor Report · Decision Letter 1]

29 Apr 2020

PONE-D-20-03246R1

Zinc transporter SLC39A7 relieves zinc deficiency to suppress alternative macrophage activation and impairment of phagocytosis

PLOS ONE

Dear Wong,

Thank you for submitting your manuscript to PLOS ONE. After careful consideration, we feel that it has merit but does not fully meet PLOS ONE’s publication criteria as it currently stands. Therefore, we invite you to submit a revised version of the manuscript that addresses the points raised during the review process.

We would appreciate receiving your revised manuscript by Jun 13 2020 11:59PM. To enhance the reproducibility of your results, we recommend that if applicable you deposit your laboratory protocols in protocols.io, where a protocol can be assigned its own identifier (DOI) such that it can be cited independently in the future. For instructions see: http://journals.plos.org/plosone/s/submission-guidelines#loc-laboratory-protocols

We look forward to receiving your revised manuscript.

Kind regards,

Rebecca A Hall

Academic Editor

PLOS ONE

Additional Editor Comments (if provided):

Thank you for submitting a revised version of your manuscript. I can see that you have addressed most of the comments made by the Reviewers but there are still a few points of clarification required before the manuscript can be submitted.

The main finding of the study appears to be that knockdown of SLC39A7 reduces phagocytosis of BCG, through the depletion of intracellular Zn levels, which potentially results in reduced expression of Clec4e. However, this is not clear from the text. Some sections of the text imply that SLC39A7 is directly involved in the phagocytosis of BGC (i.e. line 167-168), which I do not think the authors mean to imply. I think this is a communication issue, rather than an issue with the interpretation of the data, and I strongly suggest that the author’s seek help from a native speaker. Even with the additional proof reading between the original submission and the revised version many English mistakes remain that affect the understanding of the manuscript.

Fig2D: Zn does not restore adhesion in KD2, only KD4. This was raised by Reviewer 1, and addressed in the rebuttal letter, but no mention was made in the actual manuscript. An explanation in the manuscript is required. Did you measure intracellular Zn after the addition of extracellular Zn to check that intracellular Zn levels were restored in both cases?

Were the other phenotypes (cytokine expression and PRR expression) restored by the supplementation of exogenous Zn?

Fig5: why is there no data for the Clec4e control?

Line 167-168: We found that SLC39A7 mediated the phagocytosis of BCG by THP-1 cells. Do you really mean this?

Line 209: Unregulated, should this be up-regulated?

Figure 6 could be combined with Fig 3

---

## [Author Response · Author response to Decision Letter 1]

18 Jun 2020

1) Q：The main finding of the study appears to be that knockdown of SLC39A7 reduces phagocytosis of BCG, through the depletion of intracellular Zn levels, which potentially results in reduced expression of Clec4e. However, this is not clear from the text. Some sections of the text imply that SLC39A7 is directly involved in the phagocytosis of BCG (i.e. line 167-168), which I do not think the authors mean to imply. 

A：Thank for your kindly advice. The sentence “SLC39A7 mediated the phagocytosis to BCG-p” was now replaced with “SLC39A7 played a role in the phagocytosis of BCG by THP-1 cells (line 174)”.

We have edited the manuscript extensively and corrected many instances of poor usage of English. They are all highlighted in RED colors.

2) Q：Fig2D: Zn does not restore adhesion in KD2, only KD4. This was raised by Reviewer 1, and addressed in the rebuttal letter, but no mention was made in the actual manuscript. An explanation in the manuscript is required. Did you measure intracellular Zn after the addition of extracellular Zn to check that intracellular Zn levels were restored in both cases?

A: We performed new experiments and found that zinc supplementation substantially increased intracellular Zn2+ levels in KD2 and KD4 (Fig. S3). This elevated intracellular zinc level by supplementation was not sufficient to restore the adhesion defect in KD2. Thus, the adhesion defect in KD2 could be independent of a SLC39A7 defect that could be restored by zinc supplementation.

The following statements were added to explain the unresponsive phenotypes in KD2 (line 207-214 in Discussion):

“One of the KD cells KD-2 did not exhibit defects in several cytokine expressions (Fig. 4A-B), and was unresponsive zinc supplementation. We presumed that the different adhesion of the two knockdown cells caused the discrepancy. KD-4 has a bigger defect in adhesion and a more responsiveness in cytokine productions. It is interesting to note that monocytes from systemic lupus erythematosus have defects in adhesion and produce elevated levels of IL-6, TNF-a, and IL-10 than healthy monocytes [38]. Thus, the discrepancy between KD-2 and KD-4 could reflect a differential compensatory response by the KD cells.”

 Note: In the new sets of experiments only upregulation of the expression of M2 macrophage marker CD206 in response to SLC39A7 knockdown was reproduced and this upregulation could be reversed by zinc supplementation. mRNA expression of M2 macrophage marker genes Arg-1 and Ym-1 was either unchanged in knockdown cells or unresponsive to zinc supplementation. It is notable that the CD206 upregulation was not seen in KD2, consistent with its unresponsiveness. Collectively, the SLC39A7-dependent regulation of M2 macrophage marker CD206 expression (rescued by zinc supplementation) indicated that loss of SLC39A7 skewed macrophages towards alternative M2 activation.

In this second version of revised manuscript, we have one phenotype that could not be reproduced. Expression of IL-6 was down-regulated in both BCG-p infected KD cells in two of our recent new experiments, in contrast to the previous revision where expression of IL-6 was up-regulated (Fig. 5A). But since this IL-6 expression down-regulation could not be restored by zinc supplementation (Fig. 5A), this down-regulation phenotype could be unrelated to the zinc transporter function of SLC39A7. Despite this observation, the release of IL-6 proteins was dependent on the zinc transporter function of SLC39A7 (Fig. 5B), indicating that SLC39A7 likely regulates IL-6 post-translationally. Importantly, this result is consistent with the importance of SLC39A7 in supporting a proinflammatory profile for classically activated macrophage.

3) Q：Were the other phenotypes (cytokine expression and PRR expression) restored by the supplementation of exogenous Zn?

A: We also assayed the PRR (Clec4e and TLR4) mRNA expression and found that the mRNA levels of Clec4e and TLR4 restored by the supplementation of Zn2+ (line 147-149, Fig.6). We assayed the mRNA expression of cytokines (line 135-138, Fig.4, Fig.5), and found that the mRNA levels of cytokines could not restored by the supplementation of Zn2+, however the level of TNF-α and IL-6 proteins in the supernatants were restored by supplementation of exogenous Zn2+ (line 135-139, Fig.5B). IL-10 was undetectable using ELISA in uninfected and infected THP-1 cells.

4) Q：Fig5: why is there no data for the Clec4e control?

A: The expression of Clec4e in uninfected cells was negligible in comparison with that in BCG-p-infected cells where the expression of Clec4e increased dramatically, making it looks like there is no data for the Clec4e control. We reformatted the figure and included the data with the conditions of supplementation of Zn2+ (Fig.6).

5) Q：Line 167-168: We found that SLC39A7 mediated the phagocytosis of BCG by THP-1 cells. Do you really mean this?

A: We replaced the text with the following sentence “SLC39A7 played a role in the phagocytosis of BCG by THP-1 cells” (line 162).

6) Q：Line 209: Unregulated, should this be up-regulated?

A: Indeed, “up-regulated” was now used (line 207).

7) Q：Figure 6 could be combined with Fig 3

A: Fig 6 is now combined with Fig 3.

---

## [Editor Report · Decision Letter 2]

23 Jun 2020

Zinc transporter SLC39A7 relieves zinc deficiency to suppress alternative macrophage activation and impairment of phagocytosis

PONE-D-20-03246R2

Dear Dr. Wong,

We’re pleased to inform you that your manuscript has been judged scientifically suitable for publication and will be formally accepted for publication once it meets all outstanding technical requirements.

Kind regards,

Rebecca A Hall

Academic Editor

PLOS ONE

Additional Editor Comments (optional):

Please check the final version for grammatical errors.
---

## [Editor Report · Acceptance letter]

25 Jun 2020

PONE-D-20-03246R2 

Zinc transporter SLC39A7 relieves zinc deficiency to suppress alternative macrophage activation and impairment of phagocytosis. 

Dear Dr. Wong:

I'm pleased to inform you that your manuscript has been deemed suitable for publication in PLOS ONE. Congratulations! Your manuscript is now with our production department. 

Kind regards, 

on behalf of

Dr. Rebecca A Hall 

Academic Editor

PLOS ONE